# Evaluation of Antioxidant Activity, Cytotoxicity, and Genotoxicity of *Ptychotis verticillata* Essential Oil: Towards Novel Breast Cancer Therapeutics

**DOI:** 10.3390/life13071586

**Published:** 2023-07-19

**Authors:** Mohamed Taibi, Amine Elbouzidi, Sabir Ouahhoud, El Hassania Loukili, Douâae Ou-Yahya, Safae Ouahabi, Ali S. Alqahtani, Omar M. Noman, Mohamed Addi, Reda Bellaouchi, Abdeslam Asehraou, Ennouamane Saalaoui, Bouchra El Guerrouj, Khalid Chaabane

**Affiliations:** 1Laboratoire d’Amélioration des Productions Agricoles, Biotechnologie et Environnement (LAPABE), Faculté des Sciences, Université Mohammed Premier, Oujda 60000, Morocco; amine.elbouzidi@ump.ac.ma (A.E.); elguerroujb@gmail.com (B.E.G.); k.chaabane@ump.ac.ma (K.C.); 2Centre de l’Oriental des Sciences et Technologies de l’Eau et de l’Environnement (COSTEE), Université Mohammed Premier, Oujda 60000, Morocco; e.loukili@ump.ac.ma (E.H.L.); douaae.ouyahia@usmba.ac.ma (D.O.-Y.); 3Laboratory of Bioresources, Biotechnology, Ethnopharmacology and Health, Faculty of Sciences, Mohammed First University, Boulevard Mohamed VI, B.P. 717, Oujda 60000, Morocco; s.ouahhoud@ump.ac.ma (S.O.); r.bellaouchi@ump.ac.ma (R.B.); asehraou@yahoo.fr (A.A.); e.saalaoui@ump.ac.ma (E.S.); 4Laboratory of Applied and Environmental Chemistry (LCAE), Faculty of Sciences, Mohammed First University, B.P. 717, Oujda 60000, Morocco; ouahabi.safae@ump.ac.ma; 5Department of Pharmacognosy, College of Pharmacy, King Saud University, B.P. 2457, Riyadh 11451, Saudi Arabia; alalqahtani@ksu.edu.sa (A.S.A.); onoman@ksu.edu.sa (O.M.N.); 6Laboratoire de Biologie des Ligneux et des Grandes Cultures, INRAE USC1328, University of Orleans, CEDEX 2, 45067 Orléans, France

**Keywords:** breast cancer, oxidative stress, essential oil, *Ptychotis verticillata*, phytochemical characterization, antioxidant activity, genotoxicity, cytotoxicity, alternative treatment

## Abstract

Breast cancer is a disease characterized by the uncontrolled proliferation of malignant cells in breast tissue, and oxidative stress activated by an accumulation of reactive oxygen species (ROS) is associated with its development and progression. Essential oils from medicinal plants, known for their antioxidant and therapeutic properties, are being explored as alternatives. *Ptychotis verticillata*, also known as Nûnkha, is a medicinal plant native to Morocco, belonging to the Apiaceae family, and used for generations in traditional medicine. This study focuses on the phytochemical characterization of *P. verticillata* essential oil (PVEO) from the province of Oujda, Morocco, for its therapeutic properties. The essential oil was obtained by hydro-distillation, and its volatile components were analyzed by gas chromatography-mass spectrometry (GC-MS). The results revealed the presence of various aromatic and terpene compounds, with carvacrol being the most abundant compound. PVEO showed antioxidant properties in several tests, including β-carotene bleaching, 2,2-diphenyl-1-picrylhydrazyl (DPPH) scavenging and total antioxidant capacity (TAC). It also demonstrated cytotoxicity against MCF-7 and MDA-MB-231 breast cancer cell lines, with higher selectivity for MDA-MB-231. The results reveal that *Ptychotis verticillata* essential oil (PVEO) could be a promising natural alternative for the treatment of breast cancer,

## 1. Introduction

Breast cancer is characterized by the uncontrolled proliferation of malignant cells in the breast tissue. Although it can affect both sexes, it mainly affects women. According to the latest GLOBOCAN 2020 statistics, breast cancer is the fifth leading cause of cancer mortality, with around 2.3 million new cases detected worldwide [1]. In 2020, around 2.26 million breast cancer cases were reported, with a 95% uncertainty interval ranging from 2.24 to 2.79 million [2]. Oxidative stress is thought to considerably influence the development and progression of breast tumors through various mechanisms. One such mechanism involves DNA damage in breast cells, which can be attributed to oxidative stress-induced damage [1]. The rich phytochemical composition of essential oils derived from medicinal and aromatic plants in terms of secondary metabolites is mainly responsible for their antioxidant activity against free radicals and their potential in treating various diseases, including cancer [2,3].

Other secondary metabolites may have harmful or toxic biological activities. For example, some plants produce toxic secondary metabolites. These compounds may be responsible for the poisoning of certain animals, or even toxic effects on humans [4]. For this reason, it is important to assess an essential oil’s toxicity before using it. These essential oils are increasingly being explored as an alternative approach to mitigate the undesirable side effects of conventional treatments. Among these plants, *Ptychotis verticillata* Duby., commonly known as “Nûnkha” in Morocco [5], is an endemic dicotyledonous plant belonging to the Apiaceae family. It naturally grows in arid, sandy regions of eastern Morocco and is extensively utilized in traditional Moroccan medicine [6]. This plant exhibits adaptations to challenging environmental conditions, such as drought and nutrient-poor soils. Its distinctive feature is the whorled arrangement of its leaves in circles around the stem. The flowers are small and typically pale yellow.

The remarkable ability of *P. verticillata* to thrive under various stresses contributes to the abundance of secondary metabolites it produces. These secondary metabolites are bioactive molecules responsible for its diverse biological and pharmacological activities [7]. The plant is renowned for its medicinal properties, and different parts such as leaves and stems are therapeutically utilized. It is commonly employed in the treatment of digestive system disorders, particularly gastric and intestinal issues. Furthermore, it possesses analgesic, diuretic, and anti-asthmatic potentials [8]. Notably, *P. verticillata* is highly valued for its potent stimulating action, as well as its antioxidant and antimicrobial properties [5].

Essential oils are increasingly being explored as an alternative to alleviate the undesirable side effects of conventional treatments. The main objective of this study is to investigate the chemical components of *P. verticillata* essential oil (PVEO) by applying gas chromatography-mass spectrometry (GC/MS). In addition, we focus on predicting the toxicity of the essential oil using computational methods, and the genotoxicity of the essential oil was investigated using the comet assay. In addition, we evaluated the effects of the essential oil on two breast cancer cell lines, namely MCF7 and MDA-MB-231, in terms of cell toxicity. In addition, antioxidant activity was assessed utilizing the diphenyl-1-picrylhydrazyl (DPPH) scavenging assay anti-radical assay and β-Carotene Bleaching Assay and by calculating total antioxidant activity.

## 2. Materials and Methods

### 2.1. Plant Material and Essential Oil Extraction

The *Ptychotis verticillata* plant specimens were obtained from the local market in the province of Oujda, located in eastern Morocco, during the spring season of 2022. Taxonomic determination procedures were conducted, and a voucher specimen (number HUMPOM17*) was deposited at the Faculty of Sciences, Mohammed 1st University, in Oujda City, Morocco. The essential oil was extracted from the aerial part of the plant using the hydro-distillation method, employing a modified Clevenger apparatus. The device was securely sealed and positioned once it had been properly loaded with water and the plant sample. The heat source was activated and precisely set to 100 °C, ensuring a consistent and stable extraction process. Throughout the extraction, strict control of heating conditions was maintained to achieve the highest quality extraction output.

### 2.2. Qualitative and Semi-Quantitative Analysis of PVEO

Phytochemical analysis of PVEO was carried out using a Shimadzu QP2010 mass spectrometer with electron impact ionization (70 eV), in line with the technique used in our study. The NIST library of mass spectra from the GC-MS data system was used to compare results [9,10]. The peak area was used to calculate the percentage composition.

### 2.3. Antioxidant Activity

#### 2.3.1. 2,2-Diphenyl-1-picrylhydrazyl (DPPH) Scavenging Assay

The antioxidant activity of the essential oils was assessed by comparing them to a known antioxidant, ascorbic acid. Each concentration was tested in triplicate. The DPPH (2,2-diphenyl-1-picrylhydrazyl) solution was prepared at a concentration of 0.1 mM. The extract concentrations were varied (0.2, 0.4, 0.5, 0.7, 0.8, and 1 mg/mL). To initiate the test, 2.5 mL of the DPPH methanolic solution was added to 0.5 mL of each extract concentration. After incubating the mixture in the dark for 30 min, the optical density was measured at 517 nm against a blank. Ascorbic acid was employed as a positive control [11]. The scavenging activity was subsequently calculated using the provided formula:$$\mathrm{Free}\;\mathrm{Radical}\;\mathrm{Scavenging}\;\left(\%\right)=\left[\left(\frac{A_{\mathrm{blank}}-A_{\mathrm{sample}}}{A_{\mathrm{blank}}}\right)\right]\times 100$$

#### 2.3.2. β-Carotene Bleaching Assay

In order to assess the antioxidant activity of PVEO, we conducted the β-carotene bleach-based assay, utilizing the methodology described by Elbouzidi et al. (2023) [12]. This assay was employed to determine the ability of PVEO to prevent β-carotene bleaching, serving as a measure of its antioxidant potential. For accurate and reliable results, the percentage of residual color was calculated using the following formula:$$\mathrm{Residual}\;\mathrm{color}\;\left(\%\right)=\left[\left(\frac{\mathrm{Initial}\;\mathrm{OD}-\mathrm{Sample}\;\mathrm{OD}}{\mathrm{Initial}\;\mathrm{OD}}\right)\right]\times 100$$

#### 2.3.3. Total Antioxidant Capacity (TAC)

Total antioxidant activity refers to PVEO’s ability to neutralize free radicals and prevent oxidative damage in the body. It is measured in terms of vitamin C equivalent. Total antioxidant activity refers to PVEO’s ability to neutralize free radicals and prevent oxidative damage in the body. It is measured in terms of vitamin C equivalent. TAC determination in our study was carried out using the phosphorus-molybdenum technique [13], in line with the technique used in the study by Elbouzidi et al. (2023) [12]. A standard curve was generated from vitamin C and the results obtained were expressed in terms of vitamin C equivalents [14,15].

### 2.4. Prediction of the Toxicity Analysis (Pro-Tox II)

The estimation of toxicity levels was performed using the Pro-Tox II online tool (https://tox-new.charite.de/protox_II/ accessed on 21 June 2023) [16]. This tool evaluates the chemical structure of each compound present in the essential oil and compares it with a comprehensive database of known toxic compounds. Based on this analysis, the tool predicts the potential toxicity or likelihood of adverse effects in humans or other organisms. The Pro-Tox II online tool provides valuable insights into various toxicological aspects, including the lethal dose 50 (LD_50_), immunotoxicity, hepatotoxicity, mutagenicity, carcinogenicity, cytotoxicity, and toxicity class [17]. Utilizing these tools allowed for the assessment of potential therapeutic applications, as well as the identification and evaluation of toxicity risks associated with the compounds present in the *P. verticillata* essential oil (PVEO).

### 2.5. Genotoxic Effect

#### 2.5.1. Blood Sample Collection and Treatment of Cells

Blood was taken from male Wistar rats via retro-orbital collection using heparin-containing tubes, following the administration of pentobarbital anesthesia. Fresh blood (2 mL) was then combined with PBS solution lacking Ca^2+^ or Mg^2+^ ions (composed of 137 mM NaCl, 2.7 mM KCl, 10 mM Na_2_HPO_4_, 1.76 mM KH_2_PO_4_, pH 7.4) at a volume of 2 mL. The resulting diluted blood mixture was exposed to the samples under investigation. To prepare different concentrations (5, 10, 20, and 40 µg/mL) of PVEO, the oil was dissolved in dimethyl sulfoxide and added to PBS. Subsequently, 10 µL of blood cells were incubated with PVEO for two hours at a temperature of 37 degrees Celsius. The final concentration of dimethyl sulfoxide in the media did not exceed 0.2%, and the negative control was exposed to an equivalent concentration of the solvent. For the genotoxicity assay, hydrogen peroxide (250 µmol/L) was used as a positive control.

#### 2.5.2. Comet Assay

Before conducting the alkaline comet assay [18]. Slight adjustments were made to the protocol originally established by Ouahhoud et al. (2022). Following the treatment, the suspension underwent centrifugation at 4500× *g* rpm for 10 min. Once the centrifugation was complete, the supernatant was discarded, and the pellet containing the leukocytes was reconstituted in 1 mL of PBS. Three successive washes were conducted. After the final centrifugation, the pellet was dissolved in LMP agarose (0.5% *w*/*v* in PBS solution), and the resulting mixture was applied to a slide that had been pre-coated with NMP agarose (1.5% *w*/*v*). The slides were placed in the lysis solution for 5 min (2.5 M NaCl, 0.1 M Na_2_EDTA, 2 × 10^−2^ M Tris, 0.3 M NaOH, 1% Sodium N-lauroyl-sarcosine, 10% DMSO and 1% Triton X-100), followed by immersion in the dark at a temperature of 4 °C for 1 h. Following the lysis period, the slides underwent a thorough wash with double-distilled water. Subsequently, the slides were positioned on horizontally aligned gel electrophoresis equipment and a freshly prepared solution (0.3 M NaOH and 10^−3^ M Na_2_EDTA, pH = 13) was used. The DNA was then unraveled for a duration of 20 min. Migration was carried out under constant conditions of 300 mA current and 25 V voltage for a total of 20 min. It was essential to maintain a temperature of 4 °C throughout the entire migration and electrophoresis process. After migration, the slides were immersed in a neutralization buffer consisting of a Trizma solution at a concentration of 0.4 M and adjusted to a pH of 7.5 using HCl. The slides were allowed to remain in the buffer for a duration of five minutes. This neutralization process was repeated three times. After the electrophoresis and neutralization steps, the comets were visualized using the ethidium bromide method as described by Singh et al. (1988) [19].

#### 2.5.3. Microscopic Observation

The ethidium bromide-stained slides were examined and imaged using a fluorescence microscope, specifically the ZOE Cell Imager. The observation was conducted using the red channel with excitation at 556/20 nm and emission at 615/61 nm. For quantitative analysis of DNA lesions, an image analyzer coupled with Comet Assay IV image analysis software was used. This software allows for the measurement of various parameters associated with DNA lesions [20]. Fifty cells were randomly chosen from each of the two replicates per sample for analysis.

#### 2.5.4. Statistical Analysis

The statistical program GraphPad Prism 5.0 was employed to perform a one-way ANOVA for statistical data analysis. Tukey’s honest significance test was utilized to compare differences between treatment groups with a significance level set at *p* < 0.05, *p* < 0.01, *p* < 0.001, *p* < 0.0001.

### 2.6. Cytotoxic Activity against Breast Cancer Cell Lines

Assessing the effectiveness of novel natural cancer alternatives on cell lines is crucial in the research and advancement of new treatments. There are diverse methods employed to evaluate this effectiveness, with cell viability assays being the most commonly utilized, particularly the MTT test.

The MCF-7 and MDA-MB-231 cell lines are of great significance in breast cancer research. These cell lines are derived from breast tumors and exhibit distinct characteristics. The MCF-7 cell line represents hormone-dependent breast cancer, as it expresses estrogen and progesterone receptors. These cells tend to be less aggressive compared to the MDA-MB-231 cell line. The MCF-7 line is commonly employed to investigate mechanisms associated with the growth and progression of hormone-dependent tumors. On the other hand, the MDA-MB-231 cell line corresponds to triple-negative breast cancer, characterized by the absence of overexpression of estrogen, progesterone, and HER2 receptors [21].

#### 2.6.1. Cell Lines

The human mammary carcinoma cell lines MDA-MB-231 (a triple-negative breast cancer cell line lacking estrogen receptors—ER−) and MCF-7 cells (expressing estrogen receptors—ER+) were procured from the American Tissue Culture Collection (ATCC, Molsheim, France). These cell lines were cultivated under controlled conditions, including a 5% CO_2_ atmosphere and a humidified temperature of 37 °C. The cells were cultured in Roswell Park Memorial Institute (RPMI) 1640 medium, supplemented with 10% fetal calf serum (*v*/*v*) obtained from Dutscher, Brumath, France [22].

#### 2.6.2. Determination of Cell Viability 

The cells under investigation (1 × 10^4^ cells per well) were subjected to various concentrations (ranging from 6.25 to 200 μg/mL) of PVEO in a 96-well culture plate for 24, 48, and 72 h. PVEO was dissolved in dimethyl sulfoxide (DMSO) prior to exposure. Following this, the cells were treated with 20 μL of MTT reagent (0.5 mg/mL) at 37 °C for 4 h and rinsed thrice with 1× phosphate-buffered saline (PBS). Then, 150 μL of DMSO was added, and the absorbance of each sample was measured at 570 nm [23]. The experiments were replicated three times, with each experiment comprising three duplicate wells. GraphPad Prism version 8 software (GraphPad Software, La Jolla, CA, USA) was utilized to calculate the 50% inhibitory concentration (IC_50_). Cisplatin, a chemotherapeutic drug, was employed as a positive control in the study [12,22].

## 3. Results and Discussion

### 3.1. Phytochemical Composition

The volatile compounds present in *Ptychotis verticillata*, which contribute to its characteristic aroma and flavor, were subjected to analysis using gas chromatography-mass spectrometry (GC-MS). PVEO predominantly consists of terpenes and other aromatic compounds, and 23 volatile compounds were identified and quantified in PVEO, as presented in Figure 1 and Table 1. GC-MS analysis commonly reveals the presence of certain volatile compounds in *P. verticillata*. Additionally, it exhibits antioxidant, anti-inflammatory, and antifungal properties [24]. It also exhibits antioxidant, anti-inflammatory, and antifungal properties [25]. Another significant compound found in this oil is D-limonene (22.10%). γ-Terpinene (9.78%) and β-cymene (9.35%) are also identified as major terpenes in *P. verticillata* oil. α-Pinene and β-pinene, monoterpenes with a fresh, pine-like aroma, are detected as minor compounds (1.61% and 0.29%, respectively). Although several other compounds were detected in low quantities, such as (+)-2-carene, *p*-menth-1-en-4-ol, *p*-menth-1-en-8-ol, cinnamic aldehyde, and 4,4,7a-trimethyl-2,4,5,6,7,7a-hexahydro-1H-inden-1-one, among others. It is important to note that these results differ from those obtained by Taibi et al. (2023) [5]. It is important to note that these results for essential oil extracted from *P. verticillata* from the Oujda region differ from those obtained by Taibi et al. (2023) [5], who analyzed the same plant from the Touissit region of Morocco. These variations can be attributed to environmental factors, the age of the plant, the period of the vegetative cycle, genetic factors and even differences in several abiotic factors between the two harvesting sites, such as temperature, altitude, humidity, air quality, which can be influenced by factors such as air pollution, industrial emissions and meteorological conditions, and the status of Touissit as a mining town with the presence and concentration of mineral elements in the soil that can influence plant growth [26,27]. The plant obtained from Oujda was gathered at the same time as the Touissit plant, albeit with a slight disparity in size and age. This resulted in a remarkable difference in the number and percentage of bioactive molecules; the Touissit PVEO contains only 12 constituents, with thymol (37.05%), D-limonene (22.97%), γ-terpinene (15.97%), *m*-cymene (12.14%), and carvacrol (8.49%) as the main constituents [5]. It is essential to emphasize that the biological activities of essential oils are generally the result of a complex, synergistic interaction between several chemical compounds present in the essential oil rather than an isolated compound [28].

### 3.2. Antioxidant Activity

The antioxidant activity of PVEO was evaluated using the DPPH and β-Carotene bleaching tests and by determining total antioxidant capacity. The IC_50_ value, representing the concentration of antioxidants required to decrease the concentration of free radicals by 50%, was determined in order to quantitatively evaluate the antioxidant potential. As shown in Table 2, PVEO exhibited notable inhibitory activity against lipid peroxidation and demonstrated the ability to reduce the stable DPPH radical to DPPH-H, with IC_50_ values of 143.16 ± 2.17 μg/mL and 258.11 ± 2.12 μg/mL, respectively. However, these antioxidant activities were lower than those observed for the positive controls, Butylated hydroxytoluene (BHT) and ascorbic acid, with IC_50_ values ranging from 26.23 ± 5.92 μg/mL to 234.89 ± 2.36 μg/mL, respectively. These results are consistent with previous studies indicating the significant antioxidant potential of PVEO [5,29].

The antioxidant potential of essential oils is often influenced by the interaction of their major components and the synergistic effects between major and minor components [30]. PVEO is rich in bioactive molecules, particularly carvacrol and D-Limonene, which contribute to its antioxidant power [31,32]. These compounds are known to exhibit direct antioxidant activity by scavenging free radicals and converting them into stable compounds. In addition, they can act as indirect antioxidants by boosting overall antioxidant status, essentially through non-enzymatic mechanisms. Antioxidants can also act directly by recycling or regenerating other antioxidants present in the body, such as vitamin E, vitamin C and glutathione. By acting in this way, they help maintain a high antioxidant status and protect cells against oxidative damage [27]. This high antioxidant activity may be indirectly involved in anticancer activity. It neutralizes free radicals, which harm human health and can damage cells and tissues, causing diseases such as cancer. These results prompted us to evaluate the anticancer activity of PVEO against breast cancer cell lines.

### 3.3. Prediction of Organ Toxicity and Toxicity Endpoints In Silico

Table 3 presents an evaluation of the toxicological properties of 23 compounds using the Pro-Tox II system. The compounds are numbered from **1** to **23** and assessed based on several toxicological parameters, including hepatotoxicity, carcinogenicity, cytotoxicity, immunotoxicity, mutagenicity, and predicted LD_50_ (median lethal dose). Additionally, each compound is assigned a probability of active toxicity [17]. Upon analyzing the data, variations in the toxicological properties of the compounds become apparent. Compounds **1**, **2**, **6**, **8**, **13**, **18**, and **22** exhibit relatively high probabilities for hepatotoxicity, indicating a potential for liver toxicity. Carcinogenicity is indicated by compounds **3**, **4**, **7**, **11**, **17**, **21**, and **22**, with moderate to high probabilities. Cytotoxicity is generally high across most compounds, with compounds **3**, **4**, **6**, **7**, **8**, and **18** demonstrating particularly high probabilities. Immunotoxicity is evident in compounds **3**, **6**, **7**, and **13**, with moderate to high probabilities. Mutagenicity is observed in compounds **3**, **6**, **7**, **8**, **11**, **12**, and **13**, each exhibiting varying probabilities. Notably, compounds **3**, **7**, and **12** are marked with an asterisk (*) to indicate active mutagenicity. 

Compounds with higher LD_50_ values generally indicate lower toxicity. The LD_50_ values in the table range from 470 mg/kg to 5300 mg/kg. Compounds with LD_50_ values below 2000 mg/kg are classified as Class IV, indicating harmful substances if swallowed, while compounds with LD_50_ values between 2000 mg/kg and 5000 mg/kg fall under Class V, suggesting potential harm if swallowed.

Overall, the results indicate that most of the compounds are considered safe based on their toxicological evaluations. However, it is important to acknowledge that certain compounds exhibit potential risks in terms of carcinogenicity, cytotoxicity, immunotoxicity, and mutagenicity. This information is valuable for assessing the potential hazards associated with these compounds and can serve as a foundation for further research and the implementation of appropriate safety measures, particularly in drug development processes.

### 3.4. Genotoxic Effect

Genotoxicity refers to the ability of an agent to damage DNA and cause genetic mutations, which can have implications for health, including cancer development. When a genotoxic agent comes into contact with the cells of an organism, it can cause alterations in the DNA sequence, DNA strand breaks or other genetic damage. This damage can lead to mutations that can have various effects, such as the development of cancers or other genetic diseases [33]. For this reason, it is necessary to test the genotoxicity of each natural product before its pharmaceutical use.

The objective of the current study was to examine the genotoxic impact of the Essential Oil (PVEO) on rat leukocytes. The findings from our study demonstrated that the application of PVEO at concentrations ranging from 5 to 40 µg/mL did not result in any DNA damage (Figure 2). This was evident by the absence of significant changes in the percentage of DNA in the comet tail and the tail moment of DNA, when compared to the negative control. Based on these findings, PVEO can be considered non-genotoxic within the tested concentrations. It is important to emphasize that genotoxicity testing serves as an initial assessment of the safety profile for pharmaceuticals, substances, or nutraceutical [34]. The evaluation of various forms of DNA damage, encompassing single- and double-strand breaks, oxidative damage, as well as DNA-protein interactions, was carried out using the comet assay, also referred to as Single Cell Gel Electrophoresis (SCGE). This assay involved the utilization of agarose microgel electrophoresis. This well-established and precise microscopic technique is utilized for the analysis of selected prokaryotic and eukaryotic cells, both in vitro and in vivo [35,36,37,38]. In our study, we employed the alkaline method of the comet assay, which specifically detects single-strand breaks and alkalizable sites in DNA [34]. Chemical composition analysis of PVEO revealed the presence of major constituents such as carvacrol, D-limonene, beta-cymene, and terpinene. Previous studies have reported that carvacrol, at doses ranging from 81 to 810 mg/kg bw, did not induce in vivo genotoxicity or oxidative DNA damage in rat stomach and liver tissues [31]. Similarly, D-limonene, at concentrations below 10,000 µM, did not exhibit genotoxic effects on lymphocytes and V79 cells [39].

Furthermore, it has been reported that D-limonene, when used at concentrations below 10,000 µM, does not exhibit genotoxic effects on lymphocytes and V79 cells [40]. Additionally, thymol and terpinene, when present at concentrations below 0.1 mM, do not induce an increase in DNA strand breakage. However, at a higher concentration of 0.2 mM, a significant rise in DNA damage is observed. Interestingly, when thymol, terpinene, and carvacrol are present in lower quantities alongside genotoxic agents like 2-amino-3-methylimidazo [4,5-f]-quinoline (IQ) and mitomycin C (MMC), they significantly reduce the DNA strand breaks caused by these agents [41]. Considering these findings, it is plausible to suggest that PVEO may also exhibit antigenotoxic effects at appropriate concentrations.

### 3.5. The Anticancer Effect of PVEO on Breast Cancer Cell Lines

We assessed, for the first time, the anticancer potential of PVEO on two human breast cancer cell lines, namely MCF-7 and MDA-MB-231. The selectivity indexes and IC_50_ levels of PVEO were assessed and are presented in Figure 3 and Table 3. The IC_50_ value represents the concentration of PVEO required to inhibit 50% of the cancer cell growth, with the standard deviation (SD) provided to account for experimental variability (Table 4). In the MCF-7 breast cancer cell line, the IC_50_ value of PVEO was determined to be 78.44 ± 5.23 µg/mL, indicating that this concentration was necessary to achieve the desired 50% inhibition of cell growth. Similarly, for the MDA-MB-231 cell line, the IC_50_ value of PVEO was found to be 19.72 ± 4.81 µg/mL. These results highlight the potential of PVEO as an anticancer agent, particularly in the context of breast cancer treatment.

To evaluate the selectivity of PVEO towards cancer cells in comparison to non-cancerous cells, the selectivity index (SI) was calculated. The SI is determined by taking the ratio of the IC_50_ values of peripheral blood mononuclear cells (PBMC), representing non-cancerous cells, to the IC_50_ values of the tumor cells (MCF-7 and MDA-MB-231). For PVEO, the selectivity index for MCF-7 cells was calculated to be 3.97, indicating that the IC_50_ value for PBMC was approximately four times higher than that for MCF-7 cells. This suggests that PVEO exhibits some degree of selectivity towards MCF-7 cells. Similarly, for MDA-MB-231 cells, the selectivity index was determined to be 15.80, indicating a higher selectivity towards this specific breast cancer cell line. In comparison, the widely used anticancer drug cisplatin was included in the analysis for reference. The IC_50_ values for cisplatin on MCF-7 and MDA-MB-231 cells were 4.17 ± 2.38 µg/mL, and 14.30 ± 4.22 µg/mL, respectively. These results provide insight into the selectivity of PVEO and its potential as an anticancer agent, demonstrating its favorable selectivity towards breast cancer cells compared to non-cancerous cells.

The selectivity indexes for cisplatin were determined to be 7.57 for MCF-7 cells and 2.21 for MDA-MB-231 cells. Overall, these results indicate that it exhibits selectivity towards both MCF-7 and MDA-MB-231 breast cancer cell lines, with higher selectivity observed for MDA-MB-231 cells. This suggests that the essential oil of *P. verticillata* may have a greater affinity for the cellular subtype ER- (estrogen receptor negative) compared to the ER+ (estrogen receptor positive) subtype of breast cancer. The selectivity indexes further demonstrate that PVEO displays relatively higher toxicity towards cancer cells compared to non-cancerous peripheral blood mononuclear cells (PBMCs). These findings emphasize the need for further investigation into the potential therapeutic applications of PVEO as a selective treatment for breast cancer. By exhibiting selectivity towards cancer cells, PVEO holds promise as a potential targeted therapy for breast cancer, particularly in the context of ER- subtypes.

## 4. Conclusions

In conclusion, the findings of this study highlight the rich chemical composition of *Ptychotis verticillata* essential oil (PVEO), characterized by the presence of various terpenes and aromatic compounds including carvacrol, D-Limonene, γ-Terpinene, β-Cymene, and 4,4,7a-Trimethyl-2,4,5,6,7,7a-hexahydro-1H-inden-1-one. Moreover, the essential oil demonstrated strong antioxidant activity. The cytotoxicity experiments demonstrated dose-dependent impacts on both cell lines, where the MDA-MB-231 cell line exhibited heightened sensitivity. Furthermore, the results indicate that the PVEO exhibits selectivity towards both MCF-7 and MDA-MB-231 breast cancer cell lines, with a greater selectivity observed for MDA-MB-231 cells. These findings suggest that the PVEO may possess a stronger affinity for the ER- (estrogen receptor negative) cellular subtype compared to the ER+ (estrogen receptor positive) subtype of breast cancer. Importantly, no genotoxic effects were observed, and there is even evidence to suggest a potential antigenotoxic effect at certain concentrations. Based on these findings, PVEO holds promise as a natural bioactive substance for preventive and therapeutic applications, offering potential benefits without associated toxicity or adverse effects.

## Figures and Tables

**Figure 1 life-13-01586-f001:**
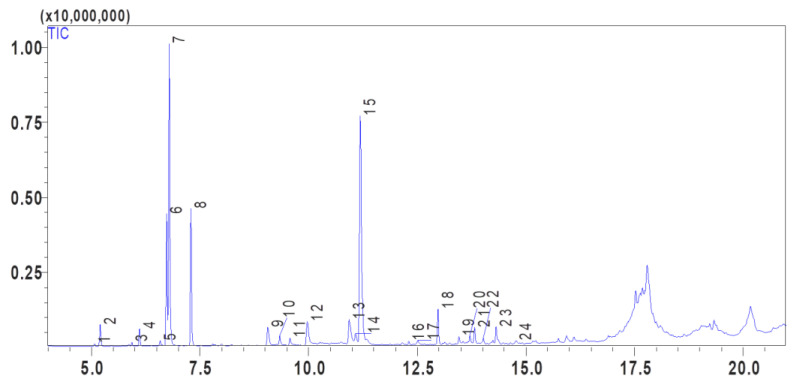
GC-MS chromatogram of *Ptychotis verticillata* essential oil (PVEO). The peaks in the chromatogram correspond to the volatile compounds identified and quantified in Table 1.

**Figure 2 life-13-01586-f002:**
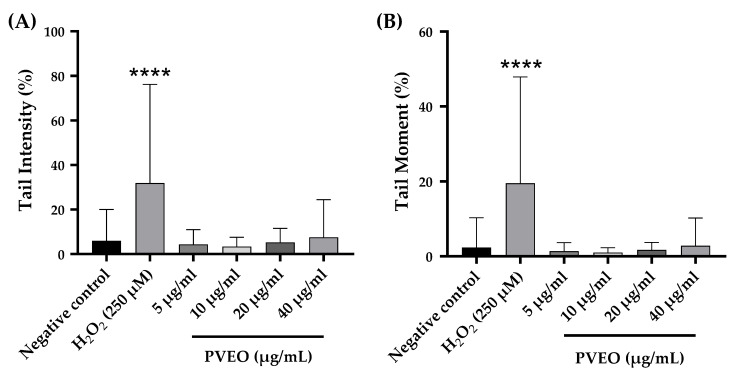
Evaluation of the effect of different concentrations of PVEO on (**A**) percentage of tail intensity and (**B**) DNA tail moment in rat leukocytes. The data are presented as mean ± SEM (50 cells × 2). **** *p* < 0.0001 compared to the negative control group.

**Figure 3 life-13-01586-f003:**
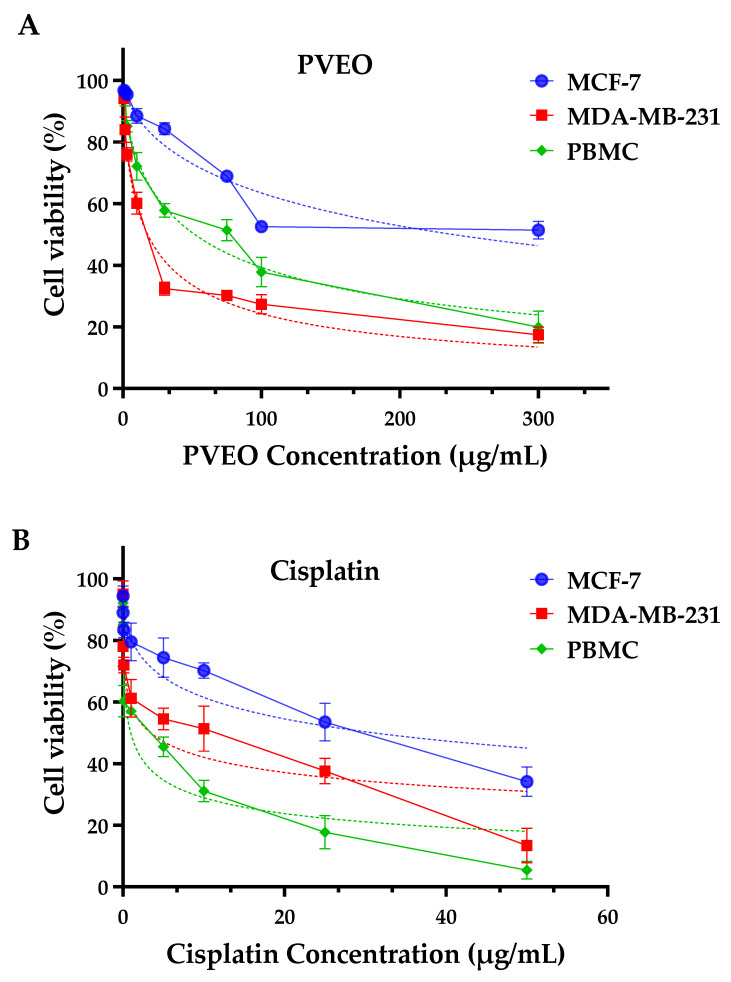
Assessment of Cell Viability in MCF-7, MDA-MB-231, and PBMC Cells Treatment with PVEO (**A**) and Cisplatin (positive control) (**B**), Following 72-h Using the MTT Assay.

**Table 1 life-13-01586-t001:** Phytochemical Composition of PVEO.

No.	Compound Name	Formula	Mol. Wt.	RT (min)	Peak Area (%)
**1**	α-Thujene	C_10_H_16_	136.23	5.075	0.21
**2**	α-Pinene	C_10_H_16_	136.23	5.204	1.61
**3**	β-Pinene	C_10_H_16_	136.23	5.933	0.29
**4**	β-Myrcene	C_10_H_16_	136.23	6.107	1.24
**5**	(+)-2-Carene	C_10_H_16_	136.23	6.582	0.60
**6**	β-Cymene	C_10_H_14_	134.22	6.732	9.35
**7**	D-Limonene	C_10_H_16_	136.23	6.792	22.10
**8**	γ-Terpinene	C_10_H_16_	136.23	7.290	9.78
**9**	*p*-Menth-4(8)-en-3-one	C_10_H_16_O	152.23	9.060	2.26
**10**	*p*-Menth-1-en-4-ol	C_10_H_18_O	154.25	9.334	0.94
**11**	*p*-Menth-1-en-8-ol	C_12_H_22_O_2_	198.30	9.568	0.93
**12**	Cinnamic aldehyde	C_9_H_8_O	132.16	9.968	4.09
**13**	4,4,7a-Trimethyl-2,4,5,6,7,7a-hexahydro-1H-inden-1-one	C_12_H_18_O	178.27	10.934	4.34
**14**	Thymol	C_10_H_14_O	150.22	11.076	1.52
**15**	Carvacrol	C_10_H_14_O	150.22	11.186	29.82
**16**	Copaene	C_15_H_24_	204.35	12.303	0.38
**17**	Caryophyllene	C_15_H_24_	204.35	12.976	3.57
**18**	1,5,9,9-Tetramethyl-1,4,7-cycloundecatriene	C_15_H_24_	204.35	13.457	0.85
**19**	Valencen	C_15_H_24_	204.35	13.710	1.16
**20**	Germacrene D	C_15_H_24_	204.35	13.817	1.56
**21**	α-Muurolene	C_15_H_24_	204.35	14.018	0.82
**22**	Δ-Cadinene	C_15_H_24_	204.35	14.313	1.96
**23**	Eicosanoic acid	C_20_H_40_O_2_	312.50	14.772	0.62
Hydrocarbon monoterpenes					45.18
Oxygenated monoterpenes					43.90
Hydrocarbon sesquiterpenes					10.30
Oxygenated sesquiterpenes					0.62
Total identified (%)					100

**Table 2 life-13-01586-t002:** Free radical scavenging and antioxidant capacity of PVEO.

EO/Reference	TAC *	β-Carotene Bleaching Assay (µg/mL)	DPPH Scavenging Capacity IC_50_ (µg/mL)
PVEO	258.98 ± 9.52	143.16 ± 2.17	258.11 ± 2.12
Ascorbic acid (AA)	-	-	234.89 ± 2.36
Butylated hydroxytoluene (BHT)	-	26.23 ± 5.92	-

* TAC expressed in µg of ascorbic acid equivalents per milligram of essential oil.

**Table 3 life-13-01586-t003:** Toxicological Evaluation of Compounds in PVEO using Pro-Tox II System. (**1**) α-Thujene, (**2**) α-Pinene, (**3**) β-Pinene, (**4**) β-Myrcene, (**5**) (+)-2-Carene, (**6**) β-Cymene, (**7**) D-Limonene, (**8**) γ-Terpinene, (**9**) *p*-Menth-4(8)-en-3-one, (**10**) *p*-Menth-1-en-4-ol, (**11**) *p*-Menth-1-en-8-ol, (**12**) Cinnamic aldehyde, (**13**) 4,4,7a-Trimethyl-2,4,5,6,7,7a-hexahydro-1H-inden-1-one, (**14**) Thymol, (**15**) Carvacrol, (**16**) Copaene, (**17**) Caryophyllene, (**18**) 1,5,9,9-Tetramethyl-1,4,7-cycloundecatriene, (**19**) Valencen, (**20**) Germacrene D, (**21**) α-Murolene, (**22**) Δ-Cadinene, (**23**) Eicosanoic acid. Note: Compounds with asterisk indicates an active toxicity.

No.	Hepatotoxicity	Carcinogenicity	Cytotoxicity	Immunotoxicity	Mutagenicity	Predicted LD_50_ (mg/kg)	Class
	Probability						
**1**	0.86	0.55	0.98	0.78	0.73	5000	V
**2**	0.86	0.60	0.99	0.93	0.75	3700	V
**3**	0.80	0.66	0.97	0.95 *	0.71 *	4700	V
**4**	0.77	0.60	0.99	0.98	0.75	5000	V
**5**	0.78	0.71	0.69	0.74	0.81	4800	V
**6**	0.87	0.67 *	0.98	0.98	0.89	2374	V
**7**	0.76	0.65	0.95	0.97 *	0.82 *	4400	V
**8**	0.83	0.60	0.98	0.92	0.82	2500	V
**9**	0.70	0.82	0.84	0.85	0.99	470	IV
**10**	0.80	0.72	0.99	0.83	0.88	1016	IV
**11**	0.72	0.76	0.90	0.64	0.99	2830	V
**12**	0.70	0.71	0.72 *	0.92	0.98	1850	IV
**13**	0.68	0.58	0.90	0.98	0.93	5000	V
**14**	0.75	0.60	0.93	0.99	0.89	640	IV
**15**	0.75	0.60	0.96	0.99	0.89	810	IV
**16**	0.97	0.57	0.85	0.78	0.99	5000	V
**17**	0.80	0.70	0.95	0.54 *	0.75	5300	V
**18**	0.82	0.76	0.79	0.97	0.87	3650	V
**19**	0.76	0.66	0.75	0.81	0.85	5000	V
**20**	0.80	0.73	0.83	0.80 *	0.87	5300	V
**21**	0.83	0.80	0.76	0.68	0.60	4400	V
**22**	0.82	0.75	0.69	0.66	0.68	4390	V
**23**	0.52	0.63	0.74	0.99	1.00	900	IV

Toxicity class explanation: IV: for substances that are harmful if swallowed (LD_50_ ranging from 300, and 2000 mg/kg), Class V: for compounds that may be harmful if swallowed (LD_50_ ranging between 2000, and 5000 mg/kg).

**Table 4 life-13-01586-t004:** Selectivity Indexes and IC_50_ Levels of PVEO on Different Human Breast Cancer Cell Lines.

Treatments	IC_50_ Value ± SD (µg/mL) *			Selectivity Index (SI) **	
	MDA-MB-231	MCF-7	PBMC	MCF-7	MDA-MB-231
PVEO	19.72 ± 4.81	78.44 ± 5.23	311.83 ± 8.21	3.97	15.80
Cisplatin	14.30 ± 4.22	4.17 ± 2.38	31.65 ± 5.41	7.57	2.21

* The mean values from three independent experiments were used for the calculation, and the results are presented as means with corresponding standard deviations. ** The selectivity index was determined by calculating the ratio of the IC_50_ values of peripheral blood mononuclear cells (PBMC) to the IC_50_ values of tumor cells.

## Data Availability

Not applicable.

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
