# Peer review of "Evaluation of Antioxidant Activity, Cytotoxicity, and Genotoxicity of Ptychotis verticillata Essential Oil: Towards Novel Breast Cancer Therapeutics"

_life, 2023, doi:10.3390/life13071586_

Round 1
Reviewer 1 Report
The paper titled: “Evaluation of Antioxidant Activity, Cytotoxicity, and Genotoxicity of Ptychotis verticillata Essential Oil: Towards Novel 3 Breast Cancer Therapeutics” submitted by the authors Taibi et al, they studied the effects of the chemical components of the essential oil of Ptychotis verticillata through the application of (GC/MS). The studied the toxicity of the essential oil using computational methods and assessed the harmful effects of the oil on two breast cancer cell lines ( MCF7 and MDA-MB-231), in terms of cellular toxicity as well as genotoxicity.
The investigation is of interest of researchers in this field. The novelty of the work is moderate.
There are some points need to be addressed before the publishing of this paper:
1. The abstract is long and need to be concise and more focused on the plant used in the study.
2. The introduction need to be rewritten. The long talk about breast cancer, oxidative stress and essential oils (paragraph1, 2 and 3) is well documented in book chapters. These paragraphs should be shortened into 4-10 sentences. The paper need to be focused on the plant used here which is P. verticillata. The work conducted on this plant and related plants in the same family should be highlighted and to be more focused. The novelty of the work should be highlighted.
3. The methods part
- The vouchering information of the P. verticillata need to be included in the first paragraph.
4. The results and discussion
- In table 1, its much better to show the retention index of detected compounds if available which is better than the retention time used. Also, compare the results from other studies if available.
- In the paragraph : 3.5. Results of the anticancer effect of PVEO on breast cancer cell lines. The discussion is completely missing.
5. In the reference list, older references than the last five years need to be removed. Recent citations are encouraged.
I give you major revision.
Author Response
Dear editors and reviewers,
We would like to express our gratitude for giving us the opportunity to improve our manuscript through the revised version, and we sincerely appreciate your valuable comments. We are particularly grateful to the reviewers for their comprehensive review, which helped improve the manuscript's quality. In response to the requested changes, we have carefully addressed each query and weakness by incorporating the suggested changes or providing a detailed response. Major changes to the revised manuscript have been highlighted in yellow for ease of identification. We assure you that all linguistic concerns and typos have been rectified in the manuscript, although they may not be mentioned explicitly in this response. Once again, we sincerely appreciate your time, effort and constructive feedback. We hope our revised manuscript successfully addresses all of the reviewers' comments and meets the necessary criteria for publication in Life.
Thank you,
kind regards
Prof. Dr. Mohamed Addi and the Co-authors
Reviewer 1
1. The abstract is long and need to be concise and more focused on the plant used in the study.
Answer: Thank you for your time used to give the insightful and vigilant comments. Changes have been made to the abstract to make it more concise, and lines (highlighted in yellow) have been added to the plant studied.
2. It is not clear how reliable your results are, especially since you write that the plant has practically not been studied. Did you study just one sample? What could be the difference in essential oil yield? In table. 2 need to add the results of statistical processing? What was used as a standard in the analysis?
Answer: We thank the reviewer for his remark. The introduction has been rewritten. The first three paragraphs have been shortened to 10 sentences, and new lines on P. verticillata have been added.
3. The vouchering information of the P. verticillata need to be included in the first paragraph.
Answer: think you, the vouchering information of P. verticillata has been included in the first paragraph. You will find it highlighted in yellow in the Word file.
4. The results and discussion
In table 1, its much better to show the retention index of detected compounds if available which is better than the retention time used. Also, compare the results from other studies if available.
Answer: Thank you for your valuable feedback regarding Table 1. We appreciate your suggestion to include the retention indices of detected compounds, as they can provide additional information beyond the retention time used. However, at this stage, we have not determined the retention indices for the compounds analyzed in our study.
In the paragraph : 3.5. Results of the anticancer effect of PVEO on breast cancer cell lines. The discussion is completely missing.
Answer: We thank the reviewer for his remark. Unfortunately, there isn't enough information on the essential oil of Ptychotis verticillata, as this plant is generally less studied than other species. In our case, we first evaluated the anticancer potential of PVEO on two human breast cancer cell lines, namely MCF-7 and MDA-MB-231. For this reason, we cannot discuss these results with other studies. This information has been added to the discussion section and highlighted in yellow.
5. In the reference list, older references than the last five years need to be removed. Recent citations are encouraged.
Answer: The list of references has been almost entirely modified, except for a few references to protocols carried out at earlier dates.

Reviewer 2 Report
The authors of a manuscript titled “Evaluation of Antioxidant Activity, Cytotoxicity, and Genotoxicity of Ptychotis verticillate Essential Oil: Towards Novel Breast Cancer Therapeutics” studied the phytochemical characterization of Ptychotis verticillate essential oil (PVEO), which a Moroccan plant. Their findings revealed that PVEO contains various aromatic and terpenes, and the authors investigated the antioxidants, cytotoxicity activity of PVEO, and toxicity of PVEO.
Overall, the manuscript needs major corrections before acceptance.
1- The authors need to extend the discussion
For example, 3.1. phytochemical composition, authors mentioned It is important to note that these results differ from those obtained by Taibi et al. (2023) [23], It should be emphasized that the composition and relative abundance of volatile compounds in P. verticillate oil can exhibit variability depending on the specific species, geographical location, plant age, vegetative cycle period, and even harvesting methods
The authors need to discuss the species, geographical location, plant age, and vegetative cycle period of PVEO used in this study and mentioned the difference from those published by Taibi et al
2- Authors in 3.2. antioxidant activity mentioned “These compounds are known to exhibit direct antioxidant activity by scavenging free radicals and converting them into stable compounds. Additionally, they can act as indirect antioxidants by enhancing the overall antioxidant status, whether through enzymatic or non-enzymatic mechanisms”
Authors need to discuss this part deeply; I suggest discussing how PVEO enhances the antioxidants status indirectly and whether this enzymatic or non-enzymatic
3- The results need to be re-organized, there are many parts in results that belong to the introduction or already known
For example, 3.3. Prediction of organ toxicity and toxicity endpoints in silico
Carcinogenicity, which signifies the potential to cause cancer
Cytotoxicity, reflecting the ability to induce cell death,
Mutagenicity, which refers to the ability to induce genetic mutations
The predicted LD50 values estimate the median lethal dose, representing the dose at
which 50% of the test subjects would be expected to die
All these definitions are not required in the discussion, authors need to remove these definitions from the results and focus on the results and how to discuss it.
4- Conclusion:
Very short and doesn’t include enough information to highlight the study. I suggest adding more results like the cytotoxicity results to the conclusion
Minor corrections:
1- Figure 3, the two graphs are cisplatin. Correct the legend of the figures. Tile of X-axis “cisplatin concentration” for both figures which is not correct. I also suggest using the log of the concentration, because it is easier to compare the cytotoxic assay results.
2- in the abstract word “ anarchical “ is an unpopular word in the scientific field, I think the authors meant uncontrolled, I suggest using a more popular scientific word, so the reader can understand the meaning
3- Also, in the introduction word “unbridled”, I suggest using more popular words
4- Page 8, line 338 “Overall, the table indicates” correct to “Overall, the result indicates not the table.
5- On page 11, line 396” Results of the anticancer effect of PVEO on breast cancer cell lines”, delete the word Results because this section is part of the results section.
6- Authors either use two digits or three digits after the decimal point, use one style throughout the manuscript
Author Response
Dear editors and reviewers,
We would like to express our gratitude for giving us the opportunity to improve our manuscript through the revised version, and we sincerely appreciate your valuable comments.
We are particularly grateful to the reviewers for their comprehensive review, which helped improve the manuscript's quality.
In response to the requested changes, we have carefully addressed each query and weakness by incorporating the suggested changes or providing a detailed response. Major changes to the revised manuscript have been highlighted in yellow for ease of identification. We assure you that all linguistic concerns and typos have been rectified in the manuscript, although they may not be mentioned explicitly in this response.
Once again, we sincerely appreciate your time, effort and constructive feedback. We hope our revised manuscript successfully addresses all of the reviewers' comments and meets the necessary criteria for publication in Life.
Thank you, kind regards
Prof. Dr. Mohamed Addi and the Co-authors
Reviewer 2 :
The authors need to extend the discussion
For example, 3.1. phytochemical composition, authors mentioned It is important to note that these results differ from those obtained by Taibi et al. (2023) [23], It should be emphasized that the composition and relative abundance of volatile compounds in P. verticillate oil can exhibit variability depending on the specific species, geographical location, plant age, vegetative cycle period, and even harvesting methods
The authors need to discuss the species, geographical location, plant age, and vegetative cycle period of PVEO used in this study and mentioned the difference from those published by Taibi et al
Answer : Thank you for your time used to give the insightful and vigilant comments, a few lines have been added to the discussion following the suggestions of reviewer 2. The geographical location, the age of the plant, the stage of the vegetative cycle, and the impact of various abiotic factors have been stated in the discussion.
- Authors in 3.2. antioxidant activity mentioned “These compounds are known to exhibit direct antioxidant activity by scavenging free radicals and converting them into stable compounds. Additionally, they can act as indirect antioxidants by enhancing the overall antioxidant status, whether through enzymatic or non-enzymatic mechanisms”
Authors need to discuss this part deeply; I suggest discussing how PVEO enhances the antioxidants status indirectly and whether this enzymatic or non-enzymatic.
Answer : This section has been expanded following suggestions from reviewer 2. Lines have been added to discuss the indirect enhancement of non-enzymatic antioxidant power.
- The results need to be re-organized, there are many parts in results that belong to the introduction or already known
For example, 3.3. Prediction of organ toxicity and toxicity endpoints in silico
Carcinogenicity, which signifies the potential to cause cancer
Cytotoxicity, reflecting the ability to induce cell death,
Mutagenicity, which refers to the ability to induce genetic mutations
The predicted LD50 values estimate the median lethal dose, representing the dose at
which 50% of the test subjects would be expected to die
All these definitions are not required in the discussion, authors need to remove these definitions from the results and focus on the results and how to discuss it.
Answer: Thank you for your fruitful remarks. The results have been reorganized in all the suggested sections, and the definitions have been removed from the discussion section.
- Conclusion:
Very short and doesn’t include enough information to highlight the study. I suggest adding more results like the cytotoxicity results to the conclusion
Answer: Thank you, the conclusion has been modified by adding the cytotoxicity conclusions.
Minor corrections:
- Figure 3, the two graphs are cisplatin. Correct the legend of the figures. Tile of X-axis “cisplatin concentration” for both figures which is not correct. I also suggest using the log of the concentration, because it is easier to compare the cytotoxic assay results.
Answer: Thank you for your constructive remarks. The correction of the figure captions has been made.

Round 2
Reviewer 1 Report
Accepted for me
Reviewer 2 Report
Thanks for corrections. Accept the manuscript in the present form.